# Indirect evidence of sex-selective abortion practices to the imbalanced sex ratio at birth in Australian migrant populations

Amanuel Tesfay Gebremedhin[1,2]*, Gizachew A. Tessema[2], Ravisha Srinivasjois[2,3], Judith A. Daire[2], Kevin A. Chai[2], Bereket Duko[2¤a], Kalayu Brhane Mruts[2¤b], Gavin Pereira[2,4]

**1** School of Nursing and Midwifery, Edith Cowan University, Perth, Australia, **2** Curtin School of Population Health, Curtin University, Perth, Australia, **3** Paediatrics and Neonatology, Joondalup Health Campus, Perth, Australia, **4** enAble Institute, Curtin University, Bentley, Western Australia, Australia

¤a Current address: Research Centre for Public Health, Equity and Human Flourishing, Torrens University, Adelaide, South Australia, Australia
¤b Current address: Murray Primary Health Network, Bendigo, Victoria, Australia
* a.gebremedhin@ecu.edu.au

## Abstract

A skewed sex ratio at birth (SRB), commonly observed in countries with high son preference can be attributed to prenatal sex-selective abortion. However, the possibility of sex selection among migrants in high-income countries has received little attention. Our study aims to identify the indirect evidence of sex-selective abortion practices to the SRB imbalance in a large Australian cohort. Our study aims to identify the indirect evidence of sex-selective abortion practices to the SRB imbalance in a large Australian cohort. In this population-based study, perinatal data were obtained from all registered births in Western Australia (WA) and New South Wales (NSW) for the period 1994 – 2015 (N = 2,175,252 births). We estimated the male-to-female sex ratio at birth (SRB) and 95% CI by mothers' country of birth stratified by sex of previous child and parity. The SRB exceeded expectations for children born to Indian, Chinese and Vietnamese mothers. For mothers from China, the SRB was 1.09 at second birth, slightly varying by sex of previous child (1.07 for male, 1.11 for a female) and markedly higher (1.34) at the third birth when the first two were female. A similar pattern was observed for Indian-born mothers. Indian and Chinese mothers had much higher induced abortion rates in early pregnancy than their Australian counterparts, which coincided with the introduction of non-invasive prenatal testing. Our study provides observational evidence that linked the male-biased SRB with prenatal sex determination followed by selective female-biased abortion practices. The findings of this study can inform public health policy decisions to address imbalanced SRB and sex-selective practices among migrants in high-income countries.

**Data availability statement:** The data that supports the findings of this study are owned by the government departments who approved the linkage and use of the data for this study. The current Human Research Ethics Committee approvals were obtained for public sharing and presentation of data on results only, meaning the unit-record level data used in this study cannot be shared by the authors. The steps involved in seeking permission for the use of the original data in this study is the same for all researchers. Researchers who wish to replicate our results can apply directly to Data Linkage, Department of Health, Western Australia, and the Centre for Health Research Linkage (CHeReL), NSW. The steps to apply for data are described at https://www.datalinkage-wa.org.au or https://www.cherel.org.au/.

**Funding:** This work was supported by funding from the National Health and Medical Research Council (Australia), including Project and Investigator Grants (#1099655 and #1173991 to GP), Investigator Grant (#1195716 to GAT), and the Western Australian Future Health Research and Innovation Fund (# WANMA/Ideas2023-24/10) to ATG. The funders had no role in the analysis, interpretations of the results, writing of the reports, and the decision to submit the paper for possible publication.

**Competing interests:** The authors have declared that no competing interests exist.

## Introduction

The "natural" sex ratio at birth (SRB) in most parts of the world is approximately 105 boys per 100 girls and usually falls between 104 and 107 with minor deviations due to infections, parental nutrition, parental age, environmental factors, hormonal changes, and other factors [1–4]. Any deviation from this natural SRB signifies some degree of sex selection, which is often influenced due to certain sex preferences [2].

Skewed SRB has been observed in several countries, particularly in China and India, with a disproportionately high number of "missing" female births, attributed to infanticide and neglect of female children [1,2]. In recent decades, sex-selective abortion has become a potential mechanism for gender-biased sex selection due to son preference, declining fertility, and easily available sex-determination technologies such as ultrasound [1,5]. Gender-biased sex selection, however, also depends on access to abortion after the first trimester, when ultrasound can first be used to detect the sex of a fetus. Previous studies have revealed the impact of ultrasound technology on sex-selective abortion in some Asian and Eastern European countries [1,2,5].

The increasing availability of non-invasive prenatal testing (NIPT), which can determine fetal sex as early as 10 weeks, has raised some ethical concerns regarding its potential misuse for sex-selective abortion. Unlike ultrasound, which detects fetal sex later in pregnancy, NIPT allows for earlier opportunities for sex selection, particularly in settings where abortion laws do not require a reason for termination. While there is no direct evidence linking NIPT to increased sex-selective abortion [6], concerns remain that its accessibility could contribute to such practices, particularly among communities with a strong son preference. However, studies indicate an increase in male-biased SRB among Asian migrants, in high-income countries such as the US [7,8], Canada [9,10], UK [11], and Australia [12,13]. Restricting the disclosure of fetal sex in NIPT results has been proposed as a feasible approach to mitigating sex-selective abortion [14]. Given projections of growing sex imbalances at birth due to pre-natal sex-selection practices [2]. It is crucial to examine how these practices may influence SRB trends, particularly in multicultural settings like Australia, where immigrants from countries with historically skewed SRB may maintain son preference and practice prenatal sex selection in their new countries of settlement.

Australia is a multicultural country with a sizeable immigrant population from Asia and other regions with a strong culture of son preference. It is plausible that such cultural preference combined with accessibility to NIPT services may explain the unbalanced SRB. Notably, in Australia, only two previous studies have examined the SRB according to the mother's country of birth [12,13], however, neither study specifically investigated sex-selective abortion practices taking into account the sex composition of older siblings and previous abortion. This study aimed to empirically identify the indirect evidence of sex-selective abortion practices for a large Australian cohort.

Global Public Health

## Materials and methods

### Study design

This was a population-based study to estimate sex ratios at birth, by parity (number of previous pregnancies), sex of the previous siblings, previous abortion, and mothers' country of birth for an Australian cohort of births in WA and NSW for the period 1994–2015.

### Data sources and study population

Perinatal records with complete or near-complete coverage were obtained from the Birth Registrations, Midwives Notifications System in Western Australia (WA), and the Perinatal Data Collection in New South Wales (NSW). These validated databases include all births of at least 20 weeks gestation or birthweight of 400 g or more if the gestational age was unknown [15,16]. Records in each jurisdiction were probabilistically linked based on maternal information to identify all births to individual women during the study period. We obtained information on infant sex, mother's country of birth, parity, birth year and induced abortion (S1 File).

### Inclusion criteria and selection process

There were 2,633,425 births during the study period. We sequentially excluded birth records for multiple gestations, missing or unspecified/indeterminate infant sex, and records with missing maternal age or parity, leaving 2,550,122 births. We restricted the cohort to those from countries with at least 20,000 births (n = 8), leaving 2,175,252 births in the primary analytic cohort. We further constructed cohorts for mothers with their first two (n = 1,084,034 pregnancies) and first three consecutive pregnancies (n = 533,376 pregnancies) during the study period (S1 Fig).

### Selection process

The final analytical sample included the country of maternal birth with at least 20,000 births during the study period (eight countries). These were Australia (n = 2,175,252), the UK (n = 102,538), New Zealand (n = 68,264), China (n = 68,003), India (n = 48,302), Vietnam (n = 41,231), Lebanon (n = 35,701), and the Philippines (n = 32,652).

### Operational definitions

**Mother's country of birth.** We classified all births based on the birth country of the mother according to the Australian Standard Classification of Countries for Social Statistics (ASCSS) 1990 [17], the Standard Australian Classification of Countries (SACC) 1998 [18], the Standard Australian Classification of Countries 2nd edition (SACC 2nd edition) [19] and the Standard Australian Classification of Countries 2011 (SACC2011) [20]. For records from NSW, we mapped the Australian Bureau of Statistics (ABS) 4-digit country of birth codes with the names according to the ABS Standard Classification of Countries. In WA, mothers' country of birth was ascertained using the mother's place of birth variable in the Birth Registration data or country of birth variable from the mother's Hospital Morbidity Data Collection (HMDC) records. The mother's place of birth variable in the Birth Registration is a free text variable including city, province, and country. We extracted the country of birth from the free text by developing a rule-based natural language processing algorithm and merged it with the MNS data [21,22]. The rule-based approach means that the algorithm is tailored to fit the dataset as opposed to applying machine learning/ statistical techniques that are more generalizable but require labelled training data which are resource and time intensive to construct. The country of birth information from the HMDC was used to ascertain the mother's place of birth and to retrieve the missing values in the Birth Registration database (i.e., birth records before 2000). Thus, the mother's country of birth was complete. In this study, the term 'mother' is used as it reflects the terminology in the original data source, which records information on the birthing parent as 'mother of the baby'. To ensure enough statistical power for computing sex ratios in the stratified analyses, we included the following

eight countries each providing at least 20,000 births over the study period: Australia, China, India, New Zealand, the UK, Vietnam, Lebanon, and the Philippines. This approach has been adopted by other studies [13,23].

**Infant sex.** Defined as the biological sex at birth, categorized as male or female.

**Parity.** Defined as the total number of previous pregnancies, categorized as (0: First birth, no previous birth; 1: Second birth, one previous birth; 2: Third birth, two previous births and 3+: Fourth or higher order births, three or more previous births).

**Time period.** The birth year was categorized into (1994–99, 2000–2004, 2005–09, and 2010–15) periods.

**The stopping rule.** A fertility decision where families chose to have no more additional children after having a desired sex composition of their children (usually after a son). Detailed operational definitions are provided in the supplementary file (S1 File).

## Statistical analysis

We used an intercept-only logistic regression model to estimate sex ratios (M/F ratios) at birth and their 95% confidence intervals (CIs). We defined deviation from the natural sex ratio as an observed SRB with a 95% CI that excludes 1.06, which is the overall SRB for Australia based on our data. We presented the overall SRB for each country and stratified by parity and sex of the previous birth(s). The conditional SRB – the SRB by sex of the previous sibling(s), was defined from two sub-cohorts: (i) mothers with their first two consecutive pregnancies during the study period (*parity 0–1 cohort),* and (ii) mothers with their first three consecutive pregnancies (*parity 0-1-2 cohort*). For the parity 0–1 cohort, we presented the sex ratio for each group (parity 0: firstborn; parity 1: second born; M: second birth, firstborn was male; F: second birth, firstborn was female). Similarly, for the parity 0-1-2 cohort, we presented the sex ratio for the following groups (parity 2: third born: no conditioning; MM: third birth, first and second birth were male; FF: Third birth, first and second births were female; Mixed: one male, one female in any order). Small cell observations were suppressed in accordance with data privacy regulations for WA Data Linkage Branch. All analyses were performed using STATA version 16.1 and data visualization using the R statistical language. The dataset was largely complete, with the proportion of missing data <0.2% for all variables (ranging from 0.1% for parity and 0.05% for infant sex).

**Sensitivity analysis.** We conducted several sensitivity analyses to ascertain the robustness of our main findings.

Firstly, we assessed the impact of the *stopping rule*, which reflects families' decision to stop having children after achieving a desired sex composition, mostly son. To assess how this influences fertility decisions and affects sex ratio of the last birth (SRLB)—as the sex ratio of the final birth when women have completed childbearing —we computed the sex-specific Parity Stopping Ratio (PSR). The sex-specific PSR represents the proportion of mothers with a given number of children who cease childbearing after having a male or female child at a specific parity. If the sex of the child does not influence fertility decisions, the sex-specific PSR should be similar, regardless of the sex of the previous birth. We reported these estimates overall and by parity. A higher SRLB compared to the general SRB suggests that the stopping rule has contributed to an imbalance in the sex ratio at birth (S2 Table). Secondly, to maximize the sample size during stratification by parity, we estimated the SRB by the sex of previous births without considering parity order (S3 Table). Finally, to check the potential influence of induced abortion and dispersion of sex-determining technology on SRB, we conducted a separate analysis in a sub-cohort of births in WA reporting the (i) number of hospitalizations with induced abortion stratified by gestational age and separation year; (ii) SRB stratified by the previous history of induced abortion; and (iii) SRB stratified by birth year (S4–S6 Tables). The hospital separation data for induced abortion was obtained from the Hospital Morbidity Data Collection in WA.

## Ethical approval

This study was approved by the Human Research Ethics Committees of the Department of Health in Western Australia and New South Wales. Each committee provided a waiver of consent for use of participants' data.

## Results

### Cohort characteristics

There were 2,175,252 births in the analytic cohort during the study period, of which 81.7% (1,778,561) were from Australian-born mothers. The youngest maternal age group (15–24 years) accounted for approximately a quarter of births to mothers from Australia and Lebanon, while advanced maternal age (>35 years) accounted for 28% of births to mothers from China and the Philippines. Parity varied considerably between maternal countries of birth. The lowest proportion of higher order parity (2 or higher) was observed in births to mothers from India (8.6%) and China (8.8%), and highest for births to mothers born in Lebanon (47%) (Table 1).

### Overall, parity-specific and maternal country-of-birth-specific sex ratios

The overall sex ratio at birth was 1.058, 95% CI 1.055–1.061. The parity-specific SRB ranged between 1.06, 95% CI 1.06–1.07 for parity 0 births to 1.05, 95% CI 1.04–1.06 for parity 3+ births. The SRB for births to Australian, UK, and New Zealand–born mothers was within the range expected for the natural sex ratio (1.05–1.07). However, the SRB exceeded the natural ratio for births to Indian (1.07, 95% CI 1.05–1.09), Chinese (1.08, 95% CI 1.07–1.10) and Vietnamese–born (1.07, 95% CI 1.05–1.09) mothers (Fig 1).

### Maternal country of birth specific sex ratios by parity

The SRB for births to Australian, UK, and New Zealand–born mothers remained relatively stable across parity. However, the SRB for births to Chinese, Indian, Vietnamese, and Philippines–born mothers increased for births with parity 0–2 with the highest SRB for births to Chinese (1.15, 95% CI 1.09–1.22) and Indian–born mothers (1.18, 95% CI 1.10–1.26) for parity 2 births (Fig 1).

### Maternal country of birth specific sex ratios by sex of the previous birth(s)

The SRB at second birth to mothers from China was 1.09, 95% CI 1.06–1.13. This ratio was elevated to 1.11, 95% CI 1.06–1.16 when the previous birth was female as compared to male (1.08, 95% CI 1.03–1.12). The SRB was markedly higher at the third birth when all previous births were females (1.34, 95% CI 1.14–1.57) than the SRB for all third births (1.11, 95% CI 1.02–1.23). The SRB at second birth to mothers from India was 1.10, 95% CI 1.06–1.15, which was elevated when the previous birth was a female (1.14, 95% CI 1.08–1.21). At the third birth, the odds of male birth were 31% higher than female birth (SRB 1.31, 95% CI 1.05–1.63) for mothers from India when all previous births were female. Similarly, the SRB at second birth when the first birth was female ranged from 1.16, 95% CI 1.08–1.24 for births to mothers from Lebanon to 1.11, 95% CI 1.04–1.20 for births to mothers from the Philippines. In contrast to mothers from China, India and Lebanon, SRBs at second and third birth to mothers from Australia, New Zealand, and the UK did not vary by sex of the previous birth (Figs 2 and 3, S1 Table).

### Sensitivity analysis

According to our sensitivity analysis, the overall Sex Ratio at Last Birth (SRLB, the sex ratio of the final birth when women have ceased their reproduction) was higher than SRB for births to mothers from China, Vietnam, Lebanon, and the Philippines indicating that the imbalanced SRB can be partly explained by stopping rule. Similarly, the PSR (proportion of women with a given number of children who cease childbearing at that specific parity) of male last births was higher than female last births across all parities for births from these countries, but most notably for births to mothers from China. For example, at parity 2 (third birth), 84.9% of mothers from China stopped childbearing after a male birth, compared to 74.4% after female birth. Consequently, the male: female PSR ratio for mothers from China was consistently above 1 at each parity, and 1.09 for all parities combined which implies that that mothers from Chinese origin are 9% more likely to stop childbearing if their last child was a son compared to a daughter. Conversely, the impact of the stopping rule - the

Table 1. Maternal characteristics of mothers giving birth in Australia (WA, NSW) by mother's country of birth, 1994–2015.

| Characteristics | | Total | Australia | China | India | New Zealand | UK | Vietnam | Lebanon | Philippines |
|---|---|---|---|---|---|---|---|---|---|---|
| | | N=2,175,252 | N=1,778,561 | N=68,003 | N=48,302 | N=68,264 | N=102,538 | N=41,231 | N=35,701 | N=32,652 |
| Maternal age | 15–24 | 446,719 (20.5) | 396,231 (22.3) | 4,110 (6.0) | 4,425 (9.2) | 15,308 (22.4) | 7,719 (7.5) | 5,214 (12.6) | 9,656 (27.0) | 4,056 (12.4) |
| | 25–29 | 638,098 (29.3) | 530,553 (29.8) | 17,846 (26.2) | 19,062 (39.5) | 17,863 (26.2) | 21,045 (20.5) | 12,447 (30.2) | 10,960 (30.7) | 8,322 (25.5) |
| | 30–34 | 682,315 (31.4) | 543,125 (30.5) | 27,068 (39.8) | 17,982 (37.2) | 20,259 (29.7) | 39,645 (38.7) | 14,086 (34.2) | 8,969 (25.1) | 11,181 (34.2) |
| | 35–39 | 338,924 (15.6) | 258,132 (14.5) | 15,325 (22.5) | 5,881 (12.2) | 12,102 (17.7) | 27,700 (27.0) | 7,799 (18.9) | 4,809 (13.5) | 7,176 (22.0) |
| | 40–44 | 65,533 (3.0) | 47,740 (2.7) | 3,487 (5.1) | 899 (1.9) | 2,607 (3.8) | 6,136 (6.0) | 1,602 (3.9) | 1,233 (3.5) | 1,829 (5.6) |
| | >=45 | 2,921 (0.1) | 2,066 (0.1) | 166 (0.2) | 52 (0.1) | 109 (0.2) | 290 (0.3) | 80 (0.2) | 73 (0.2) | 85 (0.3) |
| Parity | No previous birth | 905,703 (41.6) | 731,651 (41.1) | 36,749 (54.0) | 26,526 (54.9) | 26,657 (39.0) | 42,819 (41.8) | 18,244 (44.2) | 9,648 (27.0) | 13,409 (41.1) |
| | One previous birth | 733,523 (33.7) | 597,031 (33.6) | 25,222 (37.1) | 17,621 (36.5) | 21,426 (31.4) | 36,489 (35.6) | 15,085 (36.6) | 9,263 (25.9) | 11,386 (34.9) |
| | Two previous births | 336,346 (15.5) | 283,512 (15.9) | 4,923 (7.2) | 3,211 (6.6) | 11,006 (16.1) | 15,361 (15.0) | 5,566 (13.5) | 7,554 (21.2) | 5,213 (16.0) |
| | 3+ previous births | 199,680 (9.2) | 166,367 (9.4) | 1,109 (1.6) | 944 (2.0) | 9,175 (13.4) | 7,869 (7.7) | 2,336 (5.7) | 9,236 (25.9) | 2,644 (8.1) |
| Time period | 1994–1999 | 573,490 (26.4) | 476,940 (26.8) | 14,401 (21.2) | 4,626 (9.6) | 16,078 (23.6) | 31,640 (30.9) | 10,377 (25.2) | 11,932 (33.4) | 7,496 (23.0) |
| | 2000–2004 | 465,490 (21.4) | 390,423 (22.0) | 11,019 (16.2) | 4,449 (9.2) | 14,042 (20.6) | 20,979 (20.5) | 9,920 (24.1) | 8,322 (23.3) | 6,336 (19.4) |
| | 2005–2009 | 510,669 (23.5) | 421,794 (23.7) | 13,730 (20.2) | 10,317 (21.4) | 16,671 (24.4) | 23,156 (22.6) | 9,896 (24.0) | 7,592 (21.3) | 7,513 (23.0) |
| | 2010–2015 | 625,603 (28.8) | 489,404 (27.5) | 28,853 (42.4) | 28,910 (59.9) | 21,473 (31.5) | 26,763 (26.1) | 11,038 (26.8) | 7,855 (22.0) | 11,307 (34.6) |
| Sex* | Female | 1,056,891 (48.6) | 864,570 (48.6) | 32,658 (48.0) | 23,325 (48.3) | 33,220 (48.7) | 50,155 (48.9) | 19,892 (48.2) | 17,312 (48.5) | 15,759 (48.3) |
| | Male | 1,118,361 (51.4) | 913,991 (51.4) | 35,345 (52.0) | 24,977 (51.7) | 35,044 (51.3) | 52,383 (51.1) | 21,339 (51.8) | 18,389 (51.5) | 16,893 (51.7) |

*Sex assigned at birth.

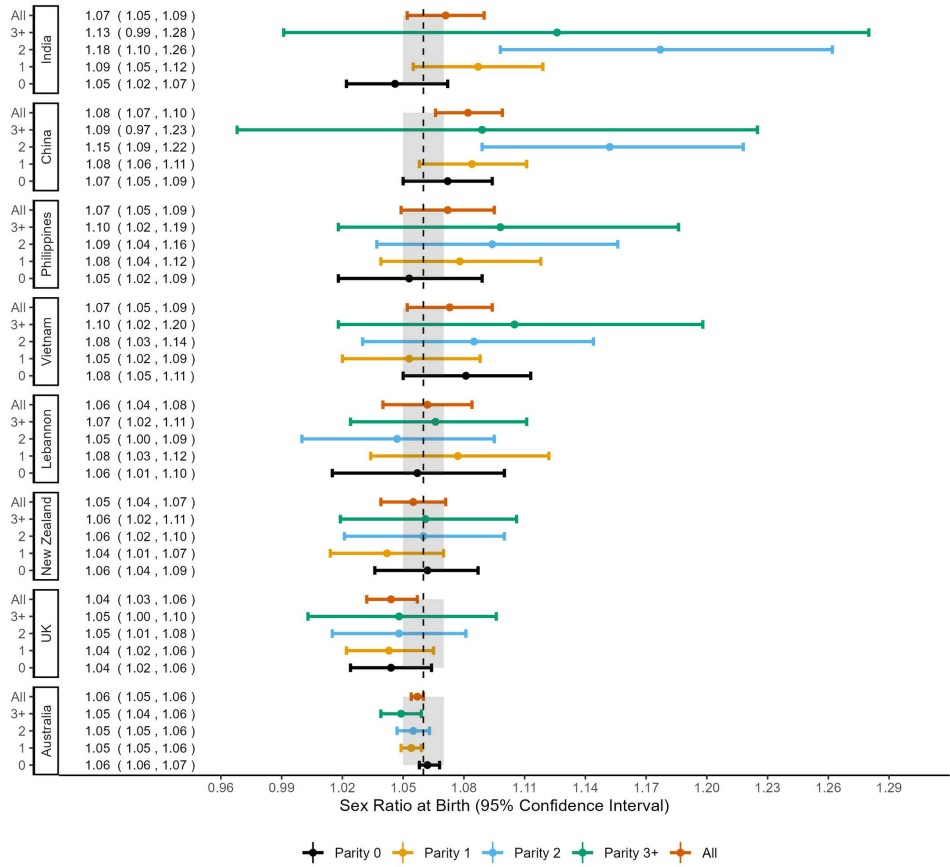

**Fig 1. Sex ratio at birth (SRB) in Australia (WA, NSW) by mother's country of birth stratified by parity, 1994–2015.** The vertical dashed line at 1.06 indicates the reference SRB, and the shaded area around it represents the 95% CI for overall SRB for Australia based on our data.

tendency of families to continue having children until they have a son - was less evident for mothers from India as only 78.6% of them stopped childbearing following male birth as compared to 75.7% following female births at parity 2 with an overall male: female PSR ratio of 0.96, indicating that they were 4% less likely to stop childbearing if their last birth was male as compared to females. Due to the smaller sample sizes, the PSR estimates for mothers born in other countries were inconsistent across parities (S2 Table). Detailed operational definitions and interpretations used in this analysis is included in the supplementary file (S1 File).

The result of our sensitivity analysis to the SRB by the sex composition without considering parity order was consistent with the main analyses supporting the hypothesis that the SRB is strongly influenced by the previous birth sex (S3 Table). During the NIPT available period (2009–2015), Indian–born mothers in WA had a higher rate of hospital induced abortions (6.7%) compared to Australian–born mothers (3.6%) with a greater proportion of early term abortions (before 13 weeks of gestation), 93% vs 84% respectively. Notably, the proportion of early abortions among Indian–born mothers increased from 64.3% in 2010 to 97.7% in 2015 (S4 Table). Our sensitivity analyses also suggested an elevated SRB for births from Indian (1.10, 95% CI 0.96–1.25) and Chinese–born (1.39, 95% CI 1.12–1.73) mothers with at least one prior abortion. Additionally, the proportion of induced abortions for Indian and Chinese–born mothers was higher if they previously had at least one daughter as compared with a previous son (S5 Table). In the year between 2010 and 2015, the SRB for births to

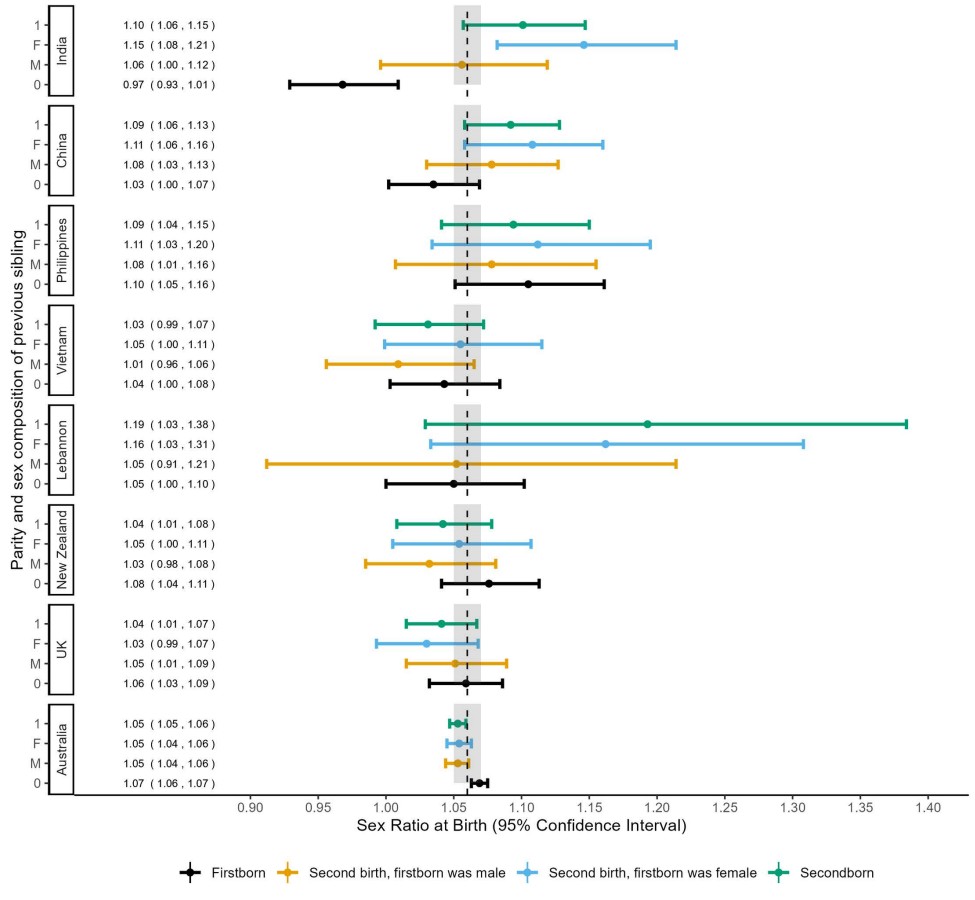

**Fig 2. Sex ratio at birth (SRB) of first order and second-order births in Australia (WA, NSW), conditional on the sex of the previous sibling for mothers with their first two consecutive births by mother's country of birth, 1994–2015.** The vertical dashed line at 1.06 indicates the reference SRB, and the shaded area around it represents the 95% CI for overall SRB for Australia based on our data.

mothers from China, India, Lebanon and the Philippines ranged from 1.06 to 1.13 and was higher in comparison to previous years prior to availability of NIPT (S6 Table).

## Discussion

Our results indicate that the SRB for Australian births to women of Asian origin, particularly from China, and India is higher than the natural ratio, especially at higher parities. We revealed male–biased SRB particularly in second and third–order births from Indian and Chinese mothers with an earlier daughter or two earlier daughters that was not explained by the stopping rule. Additionally, we observed substantially more induced abortions among Indian and Chinese-born mothers, particularly at earlier gestations, which coincided with the introduction of NIPT. This study provides the most compelling observational evidence to date of male-biased SRB among overseas-born mothers, which appears to be attributed to prenatal sex determination followed by selective abortion of females.

The study findings indicate an imbalanced SRB consistent with previous research from western countries where mothers were born overseas [7,10,24–27], including two from Australia [12,13], particularly at higher–order births.. While few studies have examined the SRB modification by previous sex [24–26], only two studies accounted for both sex of previous births

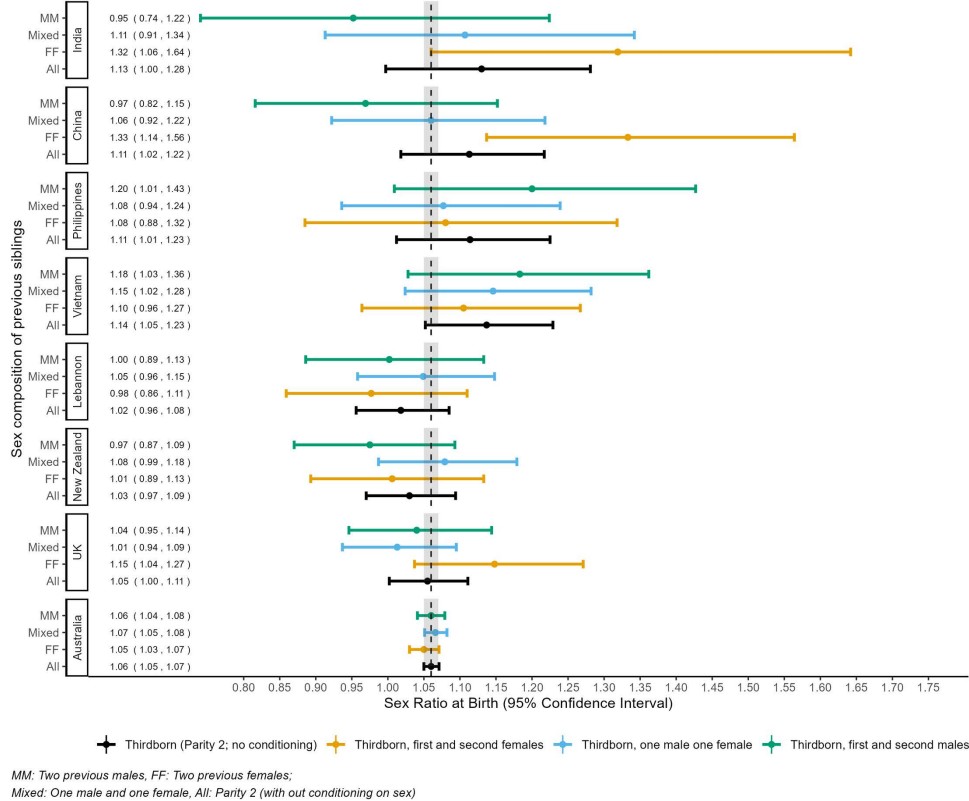

**Fig 3. Sex ratio at birth (SRB) of third-order births in Australia (WA, NSW), conditional on the sex of the previous sibling for mothers with their first three consecutive births by mother's country of birth, 1994–2015.** The vertical dashed line at 1.06 indicates the reference SRB, and the shaded area around it represents the 95% CI for overall SRB for Australia based on our data.

and previous induced abortions [9,10]. Our findings of an increased SRB when the previous births were females, as compared to male or mixed sex, align with findings of previous studies among women of Indian and Chinese origin [10,25,26,28]. Although preference for mixed–sex composition — should such a preference exist — is typical in western societies such as Australia, data on sex preference are not always unequivocal with some studies suggesting a preference for daughters [29]. Our sensitivity analyses support that the SRB is highly influenced by previous sex, particularly for births from Indian, Chinese, Vietnam and Lebanon–born mothers. Our findings of an increased SRB at parity two and male: female PSR for births from China, Lebanon, and Philippines–born mothers, suggest a preference for male children while limiting family size. Furthermore, a higher SRB for first births to Chinese and Vietnam–born mothers also indicates possible sex selection from the first birth. Although our findings suggest a consistent trend of male-biased SRB, we note that wider confidence intervals at higher order births (parity 3+) for some countries (e.g., China, India, and Vietnam) reflect smaller sample sizes and greater variability in fertility behaviors such as *stopping* rules. Despite this, the overall trend of male-biased SRB remains consistent and aligns with existing literature on sex ratio imbalances among immigrant groups, particularly from countries with a strong son preference. These findings should be interpreted with caution, particularly for higher birth orders where statistical uncertainty is greater. However, most estimates remain outside the expected natural SRB range, reinforcing the validity of our results.

Prenatal sex selection in Asian and Eastern European countries can be attributed to three major factors: son preference (*demand factor*); the effect of decreased fertility on the risk that families may end up not having a boy thereby encouraging them to resort to abortions (*the squeeze factor*); and access to prenatal sex selection technology and abortion services (*supply factor*) [5]. The demand factor posits that certain families deliberately want or avoid having daughters

or sons, which can mainly be explained in terms of current societal norms [10]. In high–income countries such as Australia with strict first–trimester abortion laws and high access to contraception [30], sex selection can occur through a differential stopping rule or sex selection via sex–selective termination of pregnancy. Our data suggest that the male–biased SRB for births to Chinese–born mothers can be partially attributed to the stopping rule, unlike for births from Indian, and other Southeast Asian–born mothers. A recent report also shows that, 97% of induced abortions in WA in the year 2009–2015 performed at <13 weeks of gestation were for reasons other than fetal abnormalities potentially including sex selection [31].

The use of sex–determining technologies, such as ultrasound has led to an increase in sex–selective abortion in Asian countries [5], and it is feasible that the ease, timing, and accuracy of NIPT could further facilitate this practice [6]. Unlike ultrasound, NIPT for determining sex is available much earlier in pregnancy, at 10 weeks of gestation, giving families ample time to seek termination services (considering the legal gestational age for termination of pregnancy in each jurisdiction). In Australia, where abortion laws do not require parents to disclose their reason for termination, restricting the disclosure of fetal sex through NIPT has been suggested as one of the most effective ways to prevent sex-selective abortion [14].

Indeed, our study shows an increase in SRB in the period of 2010–2015 and increased induced abortions at earlier gestational ages for Indian and Chinese–born mothers, around the same period that NIPT become widely available [32]. It would be beneficial to have direct evidence of sex determination through NIPT (which became available in Australia in 2012) on the incidence of induced abortion. However, we are not aware of any data or published reports on the sex ratio of aborted fetuses in a large representative cohort in Australia. Our findings suggest that male–biased SRB is more pronounced for Indian and Chinese–born mothers with prior induced abortion than their Australian counterparts indicating the impact of prenatal sex determination, including NIPT on sex–selective abortion.

Other than sex–selective abortion, neglect and infanticide of females [28], and differential under-reporting of female births are common reasons for the imbalanced SRB in Asian countries [11,28]. These factors are implausible in the context of Australia. The imbalanced SRB may be partially explained by factors including infections, maternal nutrition, and hormonal influences [2–4]. However, it is unlikely to explain the discrepancies indicated in our conditional sex ratios (conditioned on parity or previous sex). In Australia, the most likely explanation for the imbalanced SRB observed among overseas–born mothers, in our study – particularly those from India, and China is sex–selective abortion.

Abortion laws in Australia are generally decriminalized with varying restrictions on gestation and consent requirements (S2 Fig). For example, in WA, abortion was permitted up to 20 weeks, requiring approval from two doctors beyond this limit. As of March 2024, the new WA abortion reform reduced this requirement to one doctor's approval and gestational limit of 23 weeks. However, abortion after 20 weeks cannot explain the male–biased SRB in our study, as these cases would have been recorded in the MNS data with their sex information and included in our SRB estimates. The lack of reliable data on how abortion laws are implemented across jurisdictions makes it unclear whether sex-selective procedures occur within the state, interstate, or overseas. However, if sex-selective abortion does occur within the state of residence, it is more likely to happen earlier in pregnancy when such procedures are more accessible and less restrictive legal environments. While our study examines population-level SRB, individual instances of sex-selection may still occur without distorting overall result. Pre-pregnancy selection via assisted reproduction, though possible, the overall influence on population-level SRB is relatively small.

NIPT is widely available for non–medical sex determination in many countries [6,33]. But in countries such as the UK, providers are discouraged from offering it for non–medical reasons [34]. In Australia, the availability of NIPT for sex determination is widely supported [35], but guidelines proscribed sex selection for all except medical reasons, although future changes to this guideline are possible. With NIPT becoming more accessible and affordable, there may be an increased pressure to relax current guidelines, potentially leading to more sex–selective abortions and "missing" female births. Consequential demographic imbalances can lead to marital squeeze [11], reinforce gender essentialism and stereotypes, and contribute to broader societal harms, including increased violence, restrictions on women's reproductive autonomy, and negative mental health impacts [1]. These imbalances also undermine gender equality by reinforcing discriminatory norms

and limiting women's rights. This underscores the importance of ongoing SRB monitoring, enforcement of existing guidelines, and public awareness campaigns on the consequences of sex selection.

## Strengths and limitations

The study's population-based longitudinal design across a large cohort of over 2 million births, combined with the use of multi-faceted approach, allows for a thorough investigation of potential sex-selective abortion practices and contributes to a deeper understanding of male-biased sex selection. Our use of extensive data from validated population-based datasets potentially eliminates the risk of response bias. Our sensitivity analyses, such as accounting for the stopping rule and induced abortion records, confirm the consistency of the results and address potential biases, supporting our conclusions.

The study has some limitations. First, we did not examine the influence of the length of residence in Australia, as a proxy for acculturation. Nevertheless, a recent Canadian study indicates that male–biased SRB does not diminish with longer residency in the host country [26]. Additionally, similar to other large studies, accurately capturing out -of -hospital induced abortion remains a limitation, despite a comprehensive data ascertainment. Nonetheless, the elevated SRB in mothers with prior induced abortion compared to those who did not for women in the same countries for which male–biased SRB was confirmed, suggests a higher propensity for selective abortions among these mothers, thereby supporting our conclusions. We limited our analysis to maternal country of birth with at least 20,000 ensuring robust detection of male–biased SRB by parity and sex of the previous births though this restricted the range of countries analyzed. We also acknowledge that using maternal country of birth as a proxy may not fully capture the nuances of cultural and ethnic background.

Moreover, the study period (1994–2015) was determined by the availability of linked perinatal data at the time of the research. While more recent data would provide additional insights, the extensive nature of our dataset, which includes over two million births across two major Australian jurisdictions, ensures that the findings remain robust and provide valuable insights into the research question.

It is important to acknowledge that our study could not determine the specific contribution of pre-pregnancy sex selection through assisted reproduction or the direct impact of NIPT in our findings. Further research such as prospective cohort studies leveraging linked health data is warranted to establish causal pathways.

## Conclusions

Our study has implications for clinical and public health practice in Australia and possibly other high–income countries. Our findings suggest patterns consistent with male biased SRB among certain immigrant groups in Australia, particularly from countries with a strong son preference, such as India and China. While this may be indicative of prenatal sex-selection, our study does not establish causality. The availability of advanced sex–determination technologies such as NIPT has the potential to facilitate sex–selective termination of pregnancy and gives a context from which SRB monitoring needs to be further considered. To address this issue, there needs to be strict enforcement of existing legislation, culturally appropriate discussions about reproductive options, and public awareness campaigns about the consequences of sex imbalance. Furthermore, researchers are urged to try to understand how reproductive decision–making occurs in certain cultures, particularly in families from South Asian backgrounds, in a culturally acceptable manner. Finally, we strongly recommend that sex–determination technologies like NIPT should not be used to reveal non–medical traits of the fetus, including sex as restricting this information is one of the most feasible approaches to preventing sex-selective abortion and reducing gender-biased reproductive practices.

## Supporting information

**S1 File. Operational definitions and interpretations.**
(DOCX)

**S1 Fig. Selection of birth records for analysis of indirect evidence of sex-selective abortion practices to the imbalanced sex ratio at birth in Australian migrant populations (WA, NSW), 1994–2015.**
(TIF)

**S2 Fig. Policy timeline of changes in legislation for accessing abortion services in Australia, WA and NSW.**
(TIF)

**S1 Table. Male-to-female ratios of singleton births in Australia (WA, NSW) from 1994 to 2015 by mother's country of birth and stratified by sex of previous sibling(s) for mothers with their first two (or three) consecutive births.**
(DOCX)

**S2 Table. Quantifying the role of stopping rule and sex-selective abortion using sex-specific parity stopping ratio (PSR) in Australia (WA, NSW) from 1994 to 2015 by mother's country of birth.**
(DOCX)

**S3 Table. Male-to-female ratios of singleton births in Australia (WA, NSW) by mother's country of birth and stratified by sex of previous births, 1994–2015.**
(DOCX)

**S4 Table. Number and percentage of hospital separations with induced abortion stratified by gestational age and year in WA, 2009–2015.**
(DOCX)

**S5 Table. Male-to-female ratios of singleton births in Australia by mother's country of birth and previous history of induced abortion in WA, 1994–2015.**
(DOCX)

**S6 Table. Male-to-female ratios of singleton births in Australia (WA, NSW) by mother's country of birth and stratified by previous sex and birth year, 1994–2015.**
(DOCX)

## Acknowledgments

The authors would like to thank the Linkage, Data Outputs and Research Data Services Teams at the Western Australian Data Linkage Branch, and the Centre for Health Research Linkage (CHeReL), NSW. We also thank the Data custodians for the Midwives Notification System, the Western Australian Registry of Births, Deaths and Marriages, Hospital Morbidity Data Collection, and NSW Perinatal Data Collection for providing data for this project.

## Author contributions

**Conceptualization:** Amanuel Tesfay Gebremedhin, Gavin Pereira.

**Data curation:** Amanuel Tesfay Gebremedhin, Kevin A Chai.

**Formal analysis:** Amanuel Tesfay Gebremedhin.

**Funding acquisition:** Gavin Pereira.

**Investigation:** Amanuel Tesfay Gebremedhin, Gizachew A. Tessema, Ravisha Srinivasjois, Judith A. Daire, Bereket Duko, Kalayu Brhane Mruts, Gavin Pereira.

**Methodology:** Amanuel Tesfay Gebremedhin, Gavin Pereira.

**Project administration:** Gizachew A. Tessema, Gavin Pereira.

**Software:** Amanuel Tesfay Gebremedhin, Kevin A. Chai.

**Supervision:** Gavin Pereira.

**Validation:** Gavin Pereira.

**Visualization:** Amanuel Tesfay Gebremedhin.

**Writing – original draft:** Amanuel Tesfay Gebremedhin.

**Writing – review & editing:** Amanuel Tesfay Gebremedhin, Gizachew A. Tessema, Ravisha Srinivasjois, Judith A. Daire, Kevin A. Chai, Bereket Duko, Kalayu Brhane Mruts, Gavin Pereira.

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
