## [Decision Letter · Decision Letter 0]

3 Jan 2025

PGPH-D-24-01367

Indirect Evidence of Sex-Selective Abortion Practices to the Imbalanced Sex Ratio at Birth in Australian Migrant Populations

Dear Dr. GEBREMEDHIN,

Thank you for submitting your manuscript to PLOS Global Public Health. After careful consideration, we feel that it has merit but does not fully meet PLOS Global Public Health’s publication criteria as it currently stands. Therefore, we invite you to submit a revised version of the manuscript that addresses the points raised during the review process.

The manuscript has been evaluated by two reviewers, and their comments are available below.

The reviewers have raised a number of major concerns. They request improvements to the reporting of the methodological and statistical aspects of the study, clearer presentation of the statistical results, and a more cautious interpretation of the results.

Could you please carefully revise the manuscript to address all comments raised?

We look forward to receiving your revised manuscript.

Kind regards,

Helen Howard

Staff Editor

Journal Requirements:

1. Please provide an Author Summary. This should appear in your manuscript between the Abstract (if applicable) and the Introduction, and should be 150–200 words long. The aim should be to make your findings accessible to a wide audience that includes both scientists and non-scientists. Sample summaries can be found on our website under Submission Guidelines:

https://journals.plos.org/globalpublichealth/s/submission-guidelines#loc-parts-of-a-submission.

Additional Editor Comments (if provided):

Reviewers' comments:

Reviewer's Responses to Questions

**Comments to the Author**

1. Does this manuscript meet PLOS Global Public Health’s publication criteria ? Is the manuscript technically sound, and do the data support the conclusions? The manuscript must describe methodologically and ethically rigorous research with conclusions that are appropriately drawn based on the data presented.

Reviewer #1: Partly

Reviewer #2: Partly

2. Has the statistical analysis been performed appropriately and rigorously?

Reviewer #1: I don't know

Reviewer #2: Yes

3. Have the authors made all data underlying the findings in their manuscript fully available (please refer to the Data Availability Statement at the start of the manuscript PDF file)?

Reviewer #1: Yes

Reviewer #2: Yes

4. Is the manuscript presented in an intelligible fashion and written in standard English?

Reviewer #1: Yes

Reviewer #2: Yes

5. Review Comments to the Author

Reviewer #1: This is an interesting study. I am not a quantitative analyst so I cannot comment on the statistical analysis. However, I have some feedback which should hopefully improve the paper. Most points are fairly minor. The last point is the most substantial but still not onerous.

1. The authors need to explain technical terms, e.g. the stopping rule (among others).

2. If the terminology is gendered in the data (i.e. "mother" rather than "pregnant person" then it is worth stating this.

3. Just because someone is born outside Australia does not mean they are not an Australian citizen. If you want to use Australian-born v. Chinese-born as a proxy for those with Chinese heritage, you would need to note that those who are Australian-born could still have Chinese heritage. If the data includes the mothers' stated ethnicity then it would be best to use that. Otherwise, you would need to note the limitations of using these proxies.

4. It is important to note that just because the authors do not observe a skewed sex ratio among certain ethnic groups does not mean that sex selection is not occurring. i.e. It may be that a roughly equal number of parents are selecting for males as for females, hence it does not result in a skew. However, such sex selection, even if it does not result in a skew, is still problematic. For example, some prospective parents (a sizable number of whom are white) travel overseas and incur great expense to use IVF and PGD in order to select the sex of their embryos. If this occurs among white people pre-pregnancy it is plausible that it would be occurring post-pregnancy as well.

5. Explain why you have not used more recent data.

6. Explanation of line 235 for why abortion after 20 weeks cannot explain SRB, and line 238, needs to be clearer.

7. There are more harms than those listed on lines 247-248 (i.e. reinforcing gender essentialism and gender stereotypes).

8. The point made in the last line is quite important as a way of preventing sex selective abortion, but some argument for it needs to be made earlier in the piece, not just mentioned at the very end. E.g. The fact that parents can learn the sex of their fetus, have an abortion in the first trimester and do not have to give a reason for the abortion (and it is impractical to attempt to do so anyhow) means that the only feasible way to prevent sex selective abortion is to stop including this information in NIPT results. You may wish to reference Browne’s paper “Why parents should not be told the sex of their fetus” to support this point.

Reviewer #2: The analysis using an intercept-only logistic regression model to estimate sex ratios (M/F ratios) at birth and their 95% confidence intervals (CIs) is simple and easy to interpret. However, the data is basically population drawn and the overall presentation is descriptive, rather than inferential. Therefore, statements such as ,”Our findings strongly support the hypothesis of the practice of aborting females in Australia by immigrants from countries with a strong son preference, such as India and China.” , may be a bit strong in itself.

The authors use the term, ‘parity’ which most people would interpret as ‘equivalence’. The investigators do not really define, ‘parity’ up front for the reader in this context. For example, tell the reader what a ‘parity 0-1-2’ is or refer them to the appropriate appendix early on. Only after reading through the document does this become clear.

The presentation of the Figures 1,2 and 3 could be presented more clearly and interpreted more readily for the non -statistical reader. Why in these figures are there some comparatively wide confidence intervals? Is this due to the sample size or some heterogeneity factors that are not clear?

The sensitivity analysis explanation on page 10 is not well explained. The investigators state that , according to their sensitivity analyses, the impact of the stopping rule was more notable in Chinese mothers than in Indian (S2 Table). It is not that notable without some further explanation. As in previous parts of the manuscript, use the data from S2 to demonstrate for the reader why that is the case. The S1 file does well to explain the sensitivity procedure. However, as noted, data from the actual S2 table would help in this case.

The investigators made a good effort to put all this together assuming the sampling was done properly. The clarity of the statistical explanation of the results could be enhanced as noted in the examples above.

6. PLOS authors have the option to publish the peer review history of their article (what does this mean? ). If published, this will include your full peer review and any attached files.

**Do you want your identity to be public for this peer review?** For information about this choice, including consent withdrawal, please see our Privacy Policy .

Reviewer #1: No

Reviewer #2: No

---

## [Decision Letter · Decision Letter 1]

17 Apr 2025

PGPH-D-24-01367R1

Indirect Evidence of Sex-Selective Abortion Practices to the Imbalanced Sex Ratio at Birth in Australian Migrant Populations

Dear Dr. GEBREMEDHIN,

Thank you for submitting your manuscript to PLOS Global Public Health. After careful consideration, we feel that it has merit but does not fully meet PLOS Global Public Health’s publication criteria as it currently stands. Therefore, we invite you to submit a revised version of the manuscript that addresses the points raised during the review process.

**Comments from PLOS Editorial Office** : In the revised version of your manuscript, we kindly ask you to address the following minor concern noted by us: 

We have noted that in response to Reviewer 1's comment 5, ' Explain why you have not used more recent data.' , you have provided an explanation for this in the Response to Reviewers document, however, this explanation has not been integrated into the revised version of the manuscript. Therefore, we kindly ask you to update the revised version of the manuscript to provide an explanation for the use of an older dataset. Thank you for your attention to this request.'

We look forward to receiving your revised manuscript.

Kind regards,

Annesha Sil, Ph.D.

Staff Editor

PLOS 

Journal Requirements:

1. Please provide additional details regarding participant consent. In the ethics statement in the Methods and online submission information, please ensure that you have specified (1) whether consent was informed and (2) what type you obtained (for instance, written or verbal, and if verbal, how it was documented and witnessed). If your study included minors, state whether you obtained consent from parents or guardians. If the need for consent was waived by the ethics committee, please include this information.

Additional Editor Comments (if provided):

Reviewers' comments:

Reviewer's Responses to Questions

**Comments to the Author**

1. If the authors have adequately addressed your comments raised in a previous round of review and you feel that this manuscript is now acceptable for publication, you may indicate that here to bypass the “Comments to the Author” section, enter your conflict of interest statement in the “Confidential to Editor” section, and submit your "Accept" recommendation.

Reviewer #2: All comments have been addressed

2. Does this manuscript meet PLOS Global Public Health’s publication criteria ? Is the manuscript technically sound, and do the data support the conclusions? The manuscript must describe methodologically and ethically rigorous research with conclusions that are appropriately drawn based on the data presented.

Reviewer #2: (No Response)

3. Has the statistical analysis been performed appropriately and rigorously?

Reviewer #2: (No Response)

4. Have the authors made all data underlying the findings in their manuscript fully available (please refer to the Data Availability Statement at the start of the manuscript PDF file)?

Reviewer #2: (No Response)

5. Is the manuscript presented in an intelligible fashion and written in standard English?

Reviewer #2: (No Response)

6. Review Comments to the Author

Reviewer #2: (No Response)

7. PLOS authors have the option to publish the peer review history of their article (what does this mean? ). If published, this will include your full peer review and any attached files.

**Do you want your identity to be public for this peer review?** For information about this choice, including consent withdrawal, please see our Privacy Policy .

Reviewer #2: No

---

## [Editor Report · Decision Letter 2]

30 Apr 2025

Indirect Evidence of Sex-Selective Abortion Practices to the Imbalanced Sex Ratio at Birth in Australian Migrant Populations

PGPH-D-24-01367R2

Dear Dr GEBREMEDHIN,

We are pleased to inform you that your manuscript 'Indirect Evidence of Sex-Selective Abortion Practices to the Imbalanced Sex Ratio at Birth in Australian Migrant Populations' has been provisionally accepted for publication in PLOS Global Public Health.

Best regards,

Julia Robinson

Executive Editor